# Clot Waveform Analysis Demonstrates Low Blood Coagulation Ability in Patients with Idiopathic Thrombocytopenic Purpura

**DOI:** 10.3390/jcm10245987

**Published:** 2021-12-20

**Authors:** Hideo Wada, Yuhuko Ichikawa, Minoru Ezaki, Katsuya Shiraki, Isao Moritani, Yoshiki Yamashita, Takeshi Matsumoto, Masahiro Masuya, Isao Tawara, Hideto Shimpo, Motomu Shimaoka

**Affiliations:** 1Department of Laboratory and General Medicine, Mie Prefectural General Medical Center, Yokkaichi 510-8561, Japan; 2Department of Central Laboratory, Mie Prefectural General Medical Center, Yokkaichi 510-8561, Japan; ichi911239@yahoo.co.jp (Y.I.); ajbyd06188@yahoo.co.jp (M.E.); 3Department of General Medicine, Mie Prefectural General Medical Center, Yokkaichi 510-8561, Japan; katsuya-shiraki@mie-gmc.jp (K.S.); isao-moritani@mie-gmc.jp (I.M.); 4Department of Hematology and Oncology, Mie University Graduate School of medicine, Tsu 514-8507, Japan; yamayamafan4989@yahoo.co.jp (Y.Y.); itawara@clin.medic.mie-u.ac.jp (I.T.); 5Department of Transfusion Medicine and Cell Therapy, Mie University Hospital, Tsu 514-8507, Japan; matsutak@clin.medic.mie-u.ac.jp; 6Course of Nursing Science, Mie University Graduate School of Medicine, Tsu 514-8507, Japan; mmasuya@clin.medic.mie-u.ac.jp; 7Mie Prefectural General Medical Center, Yokkaichi 510-8561, Japan; hideto-shimpo@mie-gmc.jp; 8Department of Molecular Pathobiology and Cell Adhesion Biology, Mie University Graduate School of Medicine, Tsu 514-8507, Japan; motomushimaoka@gmail.com

**Keywords:** CWA, APTT, sTF/FIXa, ITP, thrombocytopenia

## Abstract

Background: Although platelets, which contain large amounts of phospholipids, play an important role in blood coagulation, there is still no routine assay to examine the effects of platelets in blood coagulation. Methods: Hemostatic abnormalities in patients with thrombocytopenia, including those with idiopathic thrombocytopenic purpura (ITP), were examined using clot wave analysis (CWA)–small-amount tissue-factor-induced FIX activation (sTF/FIXa) and thrombin time (TT). Results: Although there were no marked differences in the three parameters of activated partial thromboplastin time (APTT) between normal healthy volunteers and typical patients with ITP, the peak heights of the CWA-sTF/FIXa were markedly low in patients with ITP. The three peak times of the CWA-sTF/FIXa in patients with a platelet count of ≤8.0 × 10^10^/L were significantly longer than those in patients with a platelet count > 8.0 × 10^10^/L and the peak heights of the CWA-sTF/FIXa in patients with a platelet count of ≤8.0 × 10^10^/L were significantly lower than those in patients with >8.0 × 10^10^/L. The peak heights of the CWA-APTT in patients with ITP were significantly lower than in patients with other types of thrombocytopenia. The three peak heights of the CWA-sTF/FIXa in ITP patients were significantly lower than those in patients with other types of thrombocytopenia. The CWA-TT showed lower peak heights and longer peak times in patients with ITP in comparison to patients with other types of thrombocytopenia. Conclusions: The CWA-sTF/FIXa and CWA-TT results showed that blood coagulation is enhanced by platelets and that the blood coagulation ability in ITP patients was low in comparison to healthy volunteers and patients with other types of thrombocytopenia.

## 1. Introduction

Immune thrombocytopenia (ITP), also known as idiopathic thrombocytopenic purpura, is an autoimmune bleeding disorder that is characterized by isolated thrombocytopenia caused by increased platelet destruction and impaired platelet production. It affects approximately 1 in 20,000 people [1,2]. Patients typically present with clinically benign mucocutaneous bleeding, but morbid internal bleeding can occur. The diagnosis of ITP remains clinical and is only possible after ruling out other causes of thrombocytopenia through history and laboratory testing [1]. Although the management of ITP is usually not complicated, it is difficult to manage in some ITP patients [3]. It is general considered that routine clotting assays, such as prothrombin time (PT) and activated partial and thromboplastin time (APTT), show normal results [4]. However, platelets play an important role in the coagulation system and ITP patients may have blood coagulation abnormalities that cannot be detected by routine assays. For this reason, the small-amount tissue factor (TF)-induced FIX activation (sTF/FIXa) assay, which reflects physiological coagulation using platelet-rich plasma, was established [5].

The clot waveform analysis (CWA)-APTT method [6] is capable of measuring very low coagulation factor VIII (FVIII) activities in hemophilia A [7] and to detect early disseminated intravascular coagulation (DIC) using biphasic waveforms [8]. Furthermore, CWA-APTT [6] is reported to be useful for the diagnosis of lupus anticoagulant, liver dysfunction, DIC, hemophilia, and factor VIII inhibitor [9,10,11,12], and for the monitoring of bypass therapy in patients who are positive for FVIII inhibitor or for anticoagulant therapy [13,14,15]. The CWA-APTT [6,15] and sTF/FIXa assays [5,14] are routine assays that are able to express the peak times and peak heights of the clotting curve or its derivative curve, easily when evaluating the thrombin burst.

In the present study, a CWA using APTT, sTF/FIXa, and thrombin time (TT) was used to examine the hemostatic abnormalities of patients with thrombocytopenia to evaluate the capablity of blood coagulation in patients with ITP.

## 2. Materials and Methods

Three hundred sixty-three samples from patients managed as outpatients or inpatients from 18 August 2020 to 17 October 2021 were examined. Samples from patients treated with anticoagulants were excluded. There were 17 samples from patients with thrombocytopenia due to ITP (platelet count ≤ 8.0 × 10^10^/L), 38 from patients with other types of thrombocytopenia, and 308 from patients without thrombocytopenia. The conditions in the patients with other types of thrombocytopenia included myelodysplastic syndrome (MDS; *n* = 16), antiphospholipid syndrome (APS; *n* = 5), bone marrow suppression (*n* = 5), malignant neoplasm (*n* = 4), liver cirrhosis (*n* = 3), DIC (*n* = 2), and other disease (*n* = 3). The diagnosis of ITP in this study was performed using the following criteria—platelet count ≤ 8.0 × 10^10^/L, positive for bleeding symptoms, and the exclusion of other causes of thrombocytopenia, decreased megakaryocyte count, and family history [16]. The conditions of the patients from whom samples were obtained included hematological malignancy (*n* = 83), solid cancer (*n* = 67), liver diseases (*n* = 44), autoimmune diseases (*n* = 21), pregnancy (*n* = 20), arterial thrombosis (*n* = 18), anemia (*n* = 16), infectious disease (*n* = 11), trauma (*n* = 8), hemostatic abnormalities (*n* = 6), aortic aneurysm (*n* = 5), diabetes mellitus (*n* = 3), venous thromboembolism (*n* = 2), heart failure (*n* = 2), and other diseases (*n* = 2).

The CWA-APTT was carried out using platelet-poor plasma (PPP), APTT-SP^®^, and an ACL-TOP^®^ system (Instrumentation Laboratory, Bedford, MA, USA) [11]. The CWA-sTF/FIXa assay [14] was carried out using platelet-rich plasma (PRP) and 2000-fold diluted HemosIL RecombiPlasTin 2G (Instrumentation Laboratory). The TT [17] was measured using thrombin (Thrombin 500 units, Mochida Pharmaceutical Co., Ltd., Tokyo, Japan) 0.5 IU and an ACL-TOP^®^ system (Instrumentation Laboratory). The CWA-APTT, CWA-sTF/FIXa, and CWA-TT assays show three curves [11]. The fibrin formation (FF, navy line) curve expressed the changes in the absorbance observed when measuring blood clotting. The first derivative peak (1st DP, red line) corresponds to the velocity of coagulation, and the second derivative peak (2nd DP, light blue) corresponds to the acceleration of coagulation. The 1st DP time and height, 2nd DP time and height, and FF time and height are called the 1st DPT, 1st DPH, 2nd DPT, 2nd DPH, FFT, and FFH, respectively (Figure 1). PRP and PPP were prepared as previously reported [5].

### Statistical Analyses

An analysis using the Stat-Flex software program demostrated that the data showed a non-normal distribution. Therefore, the median (25th–75th percentile) values are shown. The statistical significance of differences between two groups was examined using the Mann–Whitney *U*-test. *p*-values of <0.05 were considered to indicate statistical significance. All statistical analyses were carried out using the Stat-Flex software program (version 6; Artec Co Ltd., Osaka, Japan).

## 3. Results

Age and platelet count were significantly lower in patients with ITP in comparison to patients with other types of thrombocytopenia. D-dimer levels and PT-INR were significantly higher in patients with other thrombocytopenia or those without thrombocytopenia in comparison to patients with ITP (Table 1).

Although there were no marked differences in the three parameters of the CWA-APTT between normal healthy volunteers and typical patients with ITP, the peak heights of the CWA-APTT were higher and the peak times were slightly longer in patients with MDS in comparison to healthy volunteers (Figure 1), whereas the peak heights of the CWA-sTF/FIXa were markedly lower in patients with ITP and markedly higher in patients with MDS in comparison to healthy volunteers. Bleeding symptoms were significant in patients with platelet counts ≤ 1.0 × 10^10^/L or a low peak height of CWA-sTF/FIXa.

The peak times of the CWA-APTT in patients with platelet counts of >8.0 × 10^10^/L were significantly longer than those in healthy volunteers (Table 2). The 1st DPH values are presented in comparison to patients with MDS in Figure 2. The 2nd DPT and 1st DPT results of the CWA-TT, CWA-APTT in patients with platelet counts of >8.0 × 10^10^/L were significantly higher than those in healthy volunteers or patients with platelet counts of <8.0 × 10^10^/L and the FFH results of the CWA-APTT in patients with platelet counts of <8.0 or >8.0 × 10^10^/L were significantly higher than those in healthy volunteers (Figure 2a). The three peak times of the CWA-sTF/FIXa in patients with platelet counts of <8.0 × 10^10^/L were significantly longer in comparison to those with platelet counts of >8.0 × 10^10^/L (Table 2). and the peak heights of the CWA-sTF/FIXa in patients with platelet counts of <8.0 × 10^10^/L were significantly lower in comparison to those with >8.0 × 10^10^/L (Figure 2b).

The peak times of the CWA-APTT in patients with ITP were significantly shorter than those in patients with other types of thrombocytopenia (Table 3). The 1st DPH and FFH results of the CWA-APTT in patients with ITP were significantly lower in comparison to patients with other types of thrombocytopenia (Figure 3a). There were no significant differences in the three peak times of the CWA-sTF/FIXa among healthy volunteers, patients with ITP, and patients with other types of thrombocytopenia (Table 3). The three peak heights of the CWA-sTF/FIXa in patients with ITP were significantly lower in comparison to healthy volunteers and patients with other types of thrombocytopenia (Figure 3b). As there were significant difference in the platelet counts of patients with ITP and other types of thrombocytopenia, the platelet counts in other types thrombocytopenia were adjusted to those in ITP, and the difference in the peak hights of APTT and sTF/FIXa between these groups were reevaluated. Three peak heights of APTT and sTF/FIXa in ITP patients were significantly lower in comaparison to those in patients with other types of thrombocytopenia (Table 4).

The CWA-TT showed lower peak heights and longer peak times in ITP patients using PRP. These were significantly longer in patients with ITP in comparison to patients with other types of thrombocytopenia (Table 5). The 2nd DPH and 1st DPH of the CWA-TT in patients using both PRP and PPP were significantly lower in patients with ITP in comparison to patients with other types of thrombocytopenia (Figure 4).

## 4. Discussion

It is generally considered that a clear correlation exists between hemostasis and thrombocytopenia. However, in some ITP patients there is no correlation between the bleeding symptoms and the extent of thrombocytopenia [18], because total hemostasis involves platelets, coagulation factors, activation factors of coagulation, cofactor for coagulation factor, vessels, and other factors. As the CWA-sTF/FIXa [5,14] and CWA-TT [17] results for patients using PRP can reflect not only the platelet counts but also thrombin bursts [19,20], these assays may be used to evaluate the clotting ability in ITP patients.

APTT is not able show the differences in coagulation ability between patients with and without thrombocytopenia sufficiently. In the CWA-sTF/FIXa analysis, the peak times and heights were significantly different in patients with thrombocytopenia, suggesting that patients with thrombocytopenia have low blood coagulation ability in addition to low platelet aggregation and adhesion ability. However, the low blood coagulation ability in patients with thrombocytopenia varied among different underlying diseases. A relationship between the platelet counts and the peak heights of CWA-sTF/FIXa was previously reported [5]. We will futher examine whether there is relationship between the platelet function or platelet abnormalities (including platelet antibodies) and CWA-sTF/FIXa. Recombinant activated factor VII was reported to be effective in patients with Glanzmann’s thrombasthenia [21]. Thrombin generation abnormalities were reported in Quebec platelet disorder [22]. FV, which is an important coagulation factor, is present in platelets [23]. Thus, coagulation abnormalities may also be a cause of bleeding in patients with ITP.

The peak times of the CWA-APTT and sTF/FIXa were unable to show coagulation abnormalities in ITP patients, suggesting that clotting time might not be important for the evaluation of coagulation abnormalities in ITP patients. However, the peak heights of CWA-APTT and sTF/FIXa in ITP patients were markedly low in comparison to those in patients with other types of thrombocytopenia, suggesting a low blood coagulation ability in ITP patients. Furthermore, elevated D-dimer levels show the hypercoagulability in other types of thrombocytopenia. Although platelet counts were significantly lower in ITP patients than in other types of thrombocytopenia, the peak heights of the CWA-sTF/FIXa in other types thrombocytopenia were significantly high, which were significantly low in ITP. Low peak heights of CWA-sTF/FIXa may predict bleeding in ITP. In vivo, the CWA demonstrated that increased platelet counts were associated with enhanced blood coagulation using PRP [5]. Although thrombosis in ITP patients is an important problem, thrombosis generally occurs in the recovery phase, such as when using the thrombopoietin-receptor agonists [24]. Microvesicle-associated thrombin generation in ITP patients was reported to be increased after the initiation of thrombopoietin receptor agonists [25]. The CWA-sTF/FIX assay is simple and easy assay and is useful for diagnosing not only ITP but also hypercoagulability [26].

One of the mechanisms underlying the association between platelets and increased blood coagulation ability may involve thrombin bursts [19]. It was reported that CWA-TT reflects a thrombin burst in the presence of platelets [17]. The peak times of CWA-TT using PRP were markedly prolonged and the peak heights of the CWA-TT using both PRP and PPP were markedly low in ITP in comparison to patients with other types of thrombocytopenia. These findings suggest that platelets enhance blood coagulation through a thrombin burst in individuals with a normal platelet count. In ITP patients with thrombocytopenia, the thrombin burst is insufficient; thus, there may be some tendency towards bleeding. In future works, we will investigate the relationship between hemostatic abnormalities in ITP and severity and responsiveness to the treatment of ITP.

## 5. Conclusions

In patients with ITP, the degree of enhancement of blood coagulation by platelets is low. One of the causes of bleeding in ITP may be a low blood coagulation ability.

## Figures and Tables

**Figure 1 jcm-10-05987-f001:**
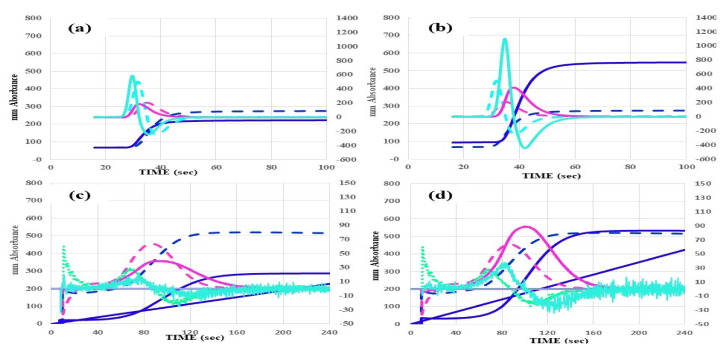
CWA-APTT (**a**,**b**) and CWA-sTF/FIXa (**c**,**d**) in patients with ITP (**a**,**c**) or MDS (**b**,**d**). CWA, clot waveform analysis; APTT, activated partial thromboplastin time; sTF/FIXa, small amount of tissue factor-induced FIX activation assay; ITP, idiopathic thrombocytopenic purpura; MDS, myelodysplastic syndrome; navy line, fibrin formation curve; red line, 1st derivative curve (velocity); light blue, 2nd derivative curve (acceleration); dashed line, normal control.

**Figure 2 jcm-10-05987-f002:**
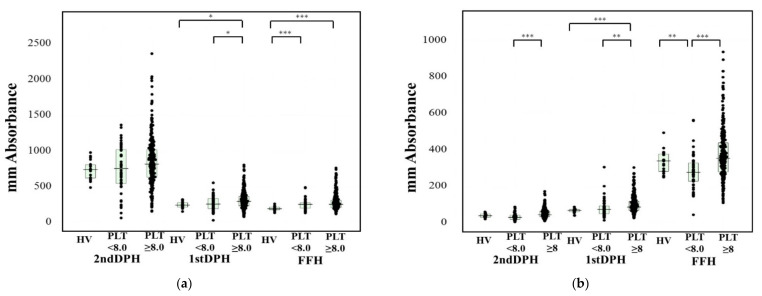
Peak heights of CWA-APTT and sTF/FIXa in healthy volunteers, patients with platelet counts of ≥8.0 × 10^4^/L, and those with <8.0 × 10^4^/L. CWA, clot waveform analysis; APTT, activated partial thromboplastin time; sTF/FIXa, small amount of tissue factor-induced FIX activation assay; HV, healthy volunteer; PLT < 8.0, platelet counts of <8.0 × 10^10^/L; PLT ≥ 8.0, platelet counts of ≥8.0 × 10^10^/L; DPH, derivative peak height; FFH, fibrin formation height; FFH, (**a**) CWA-APTT; (**b**) sTF/FIXa; *, *p* < 0.05; **, *p* < 0.01; ***, *p* < 0.001.

**Figure 3 jcm-10-05987-f003:**
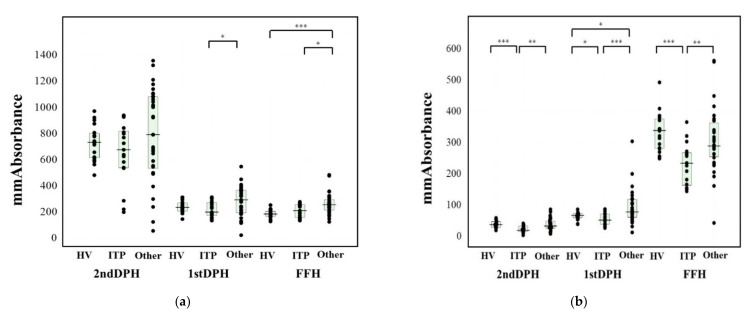
Peak heights of CWA-APTT and sTF/FIXa. CWA, clot waveform analysis; APTT, activated partial thromboplastin time; sTF/FIXa, small amount of tissue factor-induced FIX activation assay; HV, healthy volunteer; ITP, idiopathic thrombocytopenic purpura; other, other thrombocytopenia; DPH, derivative peak height; FFH, fibrin formation height; FFH, (**a**) CWA-APTT; (**b**) CWA-sTF/FIXa; *, *p* < 0.05; **, *p* < 0.01; ***, *p* < 0.001.

**Figure 4 jcm-10-05987-f004:**
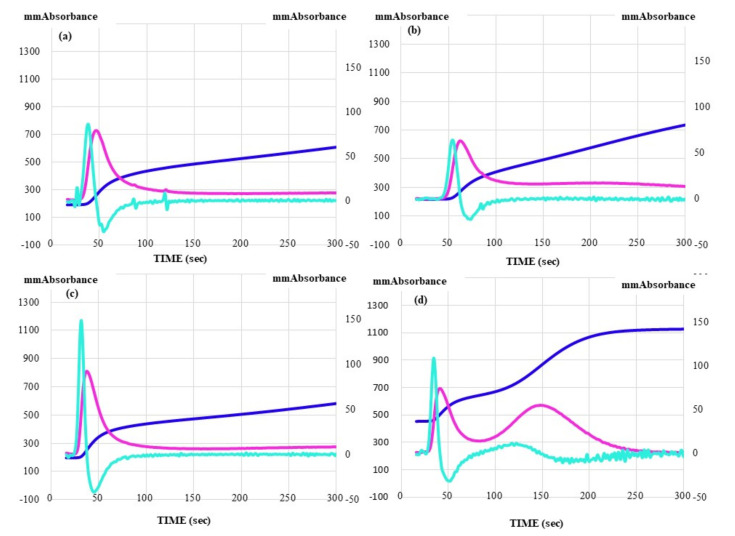
CWA-TT in PPP (**a**,**c**) and PRP (**b**,**d**) in patients with ITP (**a**,**b**) or MDS (**c**,**d**). CWA, clot waveform analysis; TT, thrombin time; PPP, platelet-poor plasma; PRP, platelet-rich plasma; ITP, idiopathic thrombocytopenic purpura; MDS, myelodysplastic syndrome; navy line, fibrin formation curve; red line, 1st derivative curve (velocity); light blue, 2nd derivative curve (acceleration).

**Table 1 jcm-10-05987-t001:** Subjects.

	*n*	Age (Years)	Sex	PLT (×10^10^/L)	D-Dimer (μg/mL)	APTT (Sec)	PT-INR
ITP	17	40 (30.0–59.0)	F, 9 M, 8	3.0 (1.1–4.2)	0.5 (0.5–0.5)	28.0 (28.0–30.3)	0.96 (0.96–0.97)
Other thrombocytopenia	38	75.0 *** (70.0–88.0)	F, 21 M, 17	5.3 ** (3.8–6.6)	1.1 *** (0.8–3.3)	33.5 (26.0–49.0)	1.07 *** (0.97–1.27)
Without thrombocytopenia	308	71.0 ***, ^###^ (57.0–80.0)	F, 115 M, 193	19.2 ***, ^###^ (14.2–25.0)	1.4 *** (0.6–4.7)	31.0 (28.0–34.0)	1.02 ** (0.97–1.13)

ITP, idiopathic thrombocytopenia; PLT, platelet count; APTT, activated partial thromboplastin time; PT-INR, prothrombin time–international normalized ratio; ***, *p* < 0.001; **, *p* < 0.01 in comparison to ITP; ^###^, *p* < 0.001 in comparison to other types of thrombocytopenia.

**Table 2 jcm-10-05987-t002:** Peak times of CWA-APTT and sTF/FIXa in healthy volunteers and patients with platelet counts of ≥8.0 × 10^4^/L or <8.0 × 10^4^/L.

	2nd DP	1st DP	FF	2nd DP	1st DP	FF
Healthy volunteers	31.7 (31.1–32.9)	34.3 (33.8–36.0)	36.0 (35.6–38.2)	77.5 (70.3–85.9)	96.5 (89.3–108)	95.4 (89.5–108)
platelet counts	CWA-APTT			CWA-sTF/FIXa		
≥8.0 × 10^4^/L	33.6 (30.7–38.5)	36.4 (33.0–40.5)	38.2 (35.2–45.9)	79.0 (74.3–88.8)	100 (90.5–113)	104 (94.1–117)
<8.0 × 10^4^/L	33.9 * (30.9–37.9)	37. 2 * (33.8–42.0)	39.3 ** (35.9–44.1)	73.6 *** (65.3–82.6)	90.4 *^###^ (81.5–102)	90.9 ^###^ (82.9–104)

Data are expressed as median (25th–75th percentile); CWA, clot waveform analysis; APTT, activated partial thromboplastin time; sTF/FIXa, small amount of tissue factor-induced FIX activation assay; 2nd DP, second derivative peak; 1st DP, first derivative peak; FF, fibrin formation; *, *p* < 0.05; **, *p* < 0.01; ***, *p* < 0.001 in comparison to healthy volunteers; ^###^, *p* < 0.001 in comparison with patients with platelet counts of <8.0 × 10^4^/L.

**Table 3 jcm-10-05987-t003:** Peak times of CWA-APTT and sTF/FIXa in healthy volunteers and patients with ITP or those with other types of thrombocytopenia.

	2nd DP	1st DP	FF	2nd DP	1st DP	FF
Healthy volunteers	31.7 (31.1–32.9)	34.3 (33.8–36.0)	36.0 (35.6–38.2)	77.5 (70.3–85.9)	96.5 (89.3–108)	95.4 (89.5–108)
	CWA-APTT			CWA-sTF/FIXa		
ITP	30.3 (29.5–33.0)	33.1 (32.0–36.4)	35.3 (33.6–38.3)	81.5 (75.1–87.5)	101 (91.2–115)	105 (97.5–121)
Other thrombocytopenia.	34.9 ***^##^ (32.6–38.8)	37.9 **^#^ (34.7–41.2)	40.6 **^#^ (36.6–47.3)	78.4 (73.7–89.3)	99.0 (90.2–113)	103 (92.7–114)

Data are expressed as median (25th–75th percentile); CWA, clot waveform analysis; APTT, activated partial thromboplastin time; sTF/FIXa, small amount of tissue factor-induced FIX activation assay; 2nd DP, second derivative peak; 1st DP, first derivative peak; FF, fibrin formation; ITP, idiopathic thrombocytopenic purpura; **, *p* < 0.01; ***, *p* < 0.001 in comparison to healthy volunteers; ^##^, *p* < 0.01; ^#^, *p* < 0.05 in comparison to ITP patients.

**Table 4 jcm-10-05987-t004:** Peak heights of CWA-APTT and sTF/FIXa in patients with ITP and patients with other types of thrombocytopenia.

	Age (Years)	APTT (mm Absorbance)			sTF/FIXa (mm Absorbance)		
		2nd DP	1st DP	FF	2nd DP	1st DP	FF
Other(*n* = 18)	3.8 (2–4.7)	679 * (541–821)	203 ** (177–275)	214 * (161–259)	21.4 * (17.7–35.4)	54.0 *** (40.5–73.7)	236 ** (165–269)
ITP (*n* = 17)	3.0 (1.1–4.2)	1010 * (670–1110)	337 ** (280–372)	257 * (239–322)	37.4 * (29.6–54.6)	97.3 *** (75.9–121)	312 ** (281–378)

Data are expressed as median (25th–75th percentile); CWA, clot waveform analysis; APTT, activated partial thromboplastin time; sTF/FIXa, small amount of tissue factor-induced FIX activation assay; 2nd DP, second derivative peak; 1st DP, first derivative peak; FF, fibrin formation; ITP, idiopathic thrombocytopenic purpura; other, other types thrombocytopenia; *, *p* < 0.05; **, *p* < 0.01; ***, *p* < 0.001 difference between ITP and other types thrombocytopenia.

**Table 5 jcm-10-05987-t005:** Peak times and heights of CWA-TT in patients with ITP and patients with other types of thrombocytopenia.

		2nd DPT	1st DPT	FFT	2nd DPH	1st DPH	FFH
ITP	PRP	41.0 * (36.3–46.8)	48.2 ** (44.5–53.7)	145 (129–167)	90.5 ** (80.3–147)	78.0 *** (75.8–96.4)	658 * (587–775)
	PPP	38.4 (36.4–41.8)	46.6 * (42.8–49.9)	189 (159–224)	114 *** (83.0–147)	90.6 *** (73.9–103)	580 * (499–737)
Other	PRP	35.0 * (32.1–40.0)	41.3 ** (38.6–46.4)	128 (113–158)	168 ** (120–272)	124 *** (101–155)	754 (697–866)
	PPP	34.5 (30.1–39.0)	40.0 * (35.5–44.9)	177 (149–194)	188 *** (152–284)	123 *** (112–171)	688 (564–758)

CWA, clot waveform; TT, thrombin time; 2nd DPT and 2nd DPH, second derivative peak time and height; 1st DPT and 1st DPH, first derivative peak time and height; FFT and FFH, fibrin formation time and height; ITP, idiopathic thrombocytopenic purpura; other, thrombocytopenia other than ITP; PRP, platelet-rich plasma; PPP, platelet-poor plasma; *, *p* < 0.05; **, *p* < 0.01; ***, *p* < 0.001 difference between ITP and other types thrombocytopenia.

## Data Availability

The data presented in this study are available on request to the corresponding author. The data are not publicly available due to privacy restrictions.

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
