# Peer review of "Clot Waveform Analysis Demonstrates Low Blood Coagulation Ability in Patients with Idiopathic Thrombocytopenic Purpura"

_jcm, 2021, doi:10.3390/jcm10245987_

Round 1
Reviewer 1 Report
Clot waveform analysis demonstrates low blood coagulation ability in patients with idiopathic thrombocytopenic purpura
The authors examined the hemostatic abnormalities of patients with thrombocytopenia by clot waveform analysis to study the blood coagulation. They found low blood coagulation ability in ITP patients.
- Statistical analyses part of the manuscript is very short and uninformative.
My first question refers to the data of the figure 2. Some data worries me about outliers, specially section a. Did the authors deal with outliers?
- The authors demonstrated their results in figure 2 and 3. It is informative to have those results in tables as supplementary material.
- The results of ITP were compared with other types of thrombocytopenia. As other types of thrombocytopenia contain different pathological conditions with different etiologies, it is advisable to have ITP results comparing with each group separately as a supplementary.
- Please rephrase these sentences: page 2, line 51 and page 7, line 222.
Author Response
Comment 1. Statistical analyses part of the manuscript is very short and uninformative.
Response 1. The statistical analysis part has now been revised.
Comment 2. My first question refers to the data of the figure 2. Some data worries me about outliers, specially section a. Did the authors deal with outliers?
Response 2. The variable data do not represent outliers or measurement errors in these assays; there are clinical reasons for these values. Several clotting assays usually result in variable data in specific conditions. We considered that variable data should not be deleted when we understood the reason for the variation.
Comment 3. The authors demonstrated their results in figure 2 and 3. It is informative to have those results in tables as supplementary material.
Response 3. The peak times of CWA-APTT and sTF/FIXa were moved to Tables 2 and 3. As the peak heights of CWA-APTT and sTF/FIXa are important, these figures have been remained as Figure 2- a and b, and 3-a and b.
Comment 4. The results of ITP were compared with other types of thrombocytopenia. As other types of thrombocytopenia contain different pathological conditions with different etiologies, it is advisable to have ITP results comparing with each group separately as a supplementary.
Response 4. Although the reviewer’s comment is reasonable, it was difficult to perform because of the small study population. Instead of this examination, we attempted another examination in Table 4 to adjust the platelet counts of patients with other types thrombocytopenia to those in patients with ITP.
Comment 5. Please rephrase these sentences: page 2, line 51 and page 7, line 222.
Response 5. These sentences have been revised.
Other major revisions.
Tables 4 and 5 have now been combined.
Highly repetitive paragraphs or sentences have now been revised.
The Manuscript has been edited by a native speaker of English.

Reviewer 2 Report
Immune thrombocytopenia is an acquired disease of platelets usually associated with low bleeding risk and mortality but often refractory to therapy and difficult to manage. Moreover, platelets may have coagulation abnormalities not highlighted by conventional routine analysis.
In this paper the authors aimed to investigated coagulation ability in 17 patients with immune thrombocytopenia using CWA-APTT, sTF/FIXa and TT and compared results with 38 patients with thrombocytopenia from other causes and 108 people with normal platelets count.
They demonstrated shorter CWA-APTT peak times and peak heights in patients with ITP compared to patients with non-immune thrombocytopenia. In ITP there were also lower CWA-sTF/FIXa. Moreover they observed a relationship between CWA-APTT and CWA-sFT/FIXa peak times an heights and platelet count (higher or lower than 8.0x1010/L).
The results is interesting and new. I have just some questions:
- From the presented data in figure 2 platelet number seemed associated with altered coagulation ability irrespective of its etiology. Please add a comment considering that in the discussion they gave a different conclusion.
- Have the authors observed differences according platelet count within the group with ITP? Is the cut off of 3.0x1010/L really protective from bleeding?
- Do the authors have any idea on the cellular cause of defective coagulation ability in ITP?
- Do they have idea if the defect is reversed by therapy? In other words did they test responsive ITP patients?
- Median age of studied ITP population is low. Did they have any idea on the coagulation defect in old people?
- How do they explain the observed lower coagulation ability in ITP and the increased thrombotic risk in primary ITP patients compared to general population? Pleas add a comment.
- Do the author suggest the routine application of the used tests?
- Finally, please check the numbers in material and methods.
Author Response
Comment 1. From the presented data in figure 2 platelet number seemed associated with altered coagulation ability irrespective of its etiology. Please add a comment considering that in the discussion they gave a different conclusion.
Response 1. In accordance with the reviewer’s suggestions, the comment has been added to the Discussion section.
Comment 2. Have the authors observed differences according platelet count within the group with ITP? Is the cut off of 3.0x1010/L really protective from bleeding?
Response 2. The relationship between bleeding symptoms and the platelet count or CWA-sTF/FIXa was added to the Result section and this point has now been discussed.
Comment 3. Do the authors have any idea on the cellular cause of defective coagulation ability in ITP?
Response 3. This point is important and has been partially discussed. Further discussion has now been added.
Comment 4. Do they have idea if the defect is reversed by therapy? In other words did they test responsive ITP patients?
Response 4. It is important, but the study population was too small to examine the relationship between the response to therapy and CWA-sTF/FIXa. This point has now been added to the Discussion section as a next step.
Comment 5. Median age of studied ITP population is low. Did they have any idea on the coagulation defect in old people?
Response 5. The relationship between age and blood coagulation abnormalities is important. However, the patients with ITP who had blood coagulation abnormalities were relatively young. Therefore, these blood coagulation abnormalities are not related to old age.
Comment 6. How do they explain the observed lower coagulation ability in ITP and the increased thrombotic risk in primary ITP patients compared to general population? Pleas add a comment.
Response 6. It is an important problem. However, the thrombotic risk in ITP increases the use of thrombopoietin analog or recovery phase of platelet counts. There are few thrombotic risks in ITP patients with a platelet count ≤8.0x1010/L in this study. This point has now been discussed.
Comment 7. Do the author suggest the routine application of the used tests?
Response 7. Yes, this assay has been carried out as a routine assay to examine not only bleeding tendency but also hypercoagulability. This point has now been discussed.
Comment 8. Finally, please check the numbers in material and methods.
Response 8. “One hundred sixty-three” was changed to “Three hundred sixty-three”.
Other major revisions.
Tables 4 and 5 have now been combined.
Highly repetitive paragraphs or sentences have now been revised.
The Manuscript has been edited by a native speaker of English.

Reviewer 3 Report
In this study, the authors use CWA with APTT, sTF/FIXa and thrombin time (TT) to evaluate hemostatic abnormalities of patients with ITP. Overall, the study is interesting and provides meaningful information. Below are few suggestions to improve the manuscript:
53: spelling error “paly”
80-81: Testing for platelet-associated immunoglobulin G positivity is not of high sensitivity and does not really correlate with clinical outcome. This is not considered as routinely indicated for diagnosis of ITP. Was this considered a required criteria for diagnosis of ITP in this study?
136-140: “The 1st DPH of the CWA-APTT in patients with platelet counts of >8.0x1010/L was significantly higher than that in healthy volunteers ..… (Figure 2b).”- How do you explain results in ITP compared to healthy volunteers?
145-152 (and statement in abstract reflecting same): Are the results in Figure 3a, 3c and 3d is just reflective of difference in platelet count in ITP (significantly lower) vs other causes of thrombocytopenia? You will need an analysis comparing results from patients with comparable platelet count, before making this statement. This is also important for the overall conclusion statement of the manuscript.
Author Response
Comment 1. 53: spelling error “paly”
Response 1. The spelling error has now been corrected.
Comment 2. 80-81: Testing for platelet-associated immunoglobulin G positivity is not of high sensitivity and does not really correlate with clinical outcome. This is not considered as routinely indicated for diagnosis of ITP. Was this considered a required criteria for diagnosis of ITP in this study?
Response 2. PAIgG was deleted. One paper that was based on our diagnosis of ITP has been cited.
Comment 3.136-140: “The 1st DPH of the CWA-APTT in patients with platelet counts of >8.0x1010/L was significantly higher than that in healthy volunteers ..… (Figure 2b).”- How do you explain results in ITP compared to healthy volunteers?
Response 3. There were no significant differences in any CWA-APTT parameters between ITP patients and healthy volunteers. I did not show the symbols for “no significance”, as I was concerned that the figures would appear too busy.
Comment 4.145-152 (and statement in abstract reflecting same): Are the results in Figure 3a, 3c and 3d is just reflective of difference in platelet count in ITP (significantly lower) vs other causes of thrombocytopenia? You will need an analysis comparing results from patients with comparable platelet count, before making this statement. This is also important for the overall conclusion statement of the manuscript.
Response 4. I agree with reviewer’s comments. We have added the reexamination data in Table 4.
Round 2
Reviewer 1 Report
Thank you for your revision.